# Combine with RNA-seq Reveals the Effect of Melatonin in the Synthesis of Melanin in Primary Melanocytes of Silky Fowls Black-Bone Chicken

**DOI:** 10.3390/genes14081648

**Published:** 2023-08-18

**Authors:** Ting Yang, Lingling Qiu, Shihao Chen, Zhixiu Wang, Yong Jiang, Hao Bai, Yulin Bi, Guobin Chang

**Affiliations:** 1Key Laboratory for Animal Genetics & Molecular Breeding of Jiangsu Province, College of Animal Science and Technology, Yangzhou University, Yangzhou 225009, China; yt1144736744@163.com (T.Y.); 007621@yzu.edu.cn (L.Q.); wangzx@yzu.edu.cn (Z.W.); jiangyong12126@163.com (Y.J.); ylbi@yzu.edu.cn (Y.B.); 2Institute of Epigenetics and Epigenomics, College of Animal Science and Technology, Yangzhou University, Yangzhou 225009, China; mrrchen@yzu.edu.cn; 3Joint International Research Laboratory of Agriculture and Agri-Product Safety, The Ministry of Education of China, Institutes of Agricultural Science and Technology Development, Yangzhou University, Yangzhou 225009, China; bhowen1027@yzu.edu.cn

**Keywords:** chicken, melanocytes, melatonin, melanin synthesis

## Abstract

(1) Background: It was found that the melanin of black-bone chicken has various effects such as scavenging DPPH free radicals and anti-oxidation, and the synthesis of melanin is affected by various factors including hormones. In addition, several studies have found that melatonin affects the melanoma cell synthesis of melanin, which has not been reported in chicken primary melanocytes; so, relevant studies were conducted. (2) Methods: In this study, chicken primary melanocytes were isolated and characterized, and then melanocytes were treated with different concentrations of melatonin to investigate the effects of melatonin on melanin synthesis in chicken melanocytes in terms of melanin synthesis-related genes, melanin content, and tyrosinase activity, and combined with RNA seq to detect the change in gene expression level of chicken melanocytes after melatonin treatment. (3) Results: We isolated and characterized primary melanocytes, and indirect immunofluorescence assay results showed positive melanocyte marker genes. RT-qPCR results showed that melatonin decreased the expression of melanin synthesis-related genes. In addition, melatonin reduced the melanin content and decreased the tyrosinase activity of melanocytes in the treated group. A total of 1703 differentially expressed genes were screened by RNA-seq, and in addition, in the KEGG results, the signaling pathway associated with melanin synthesis, and the mTOR signaling pathway were enriched. (4) Conclusions: Melatonin could decrease the synthesis of melanin in chicken primary melanocytes.

## 1. Introduction

Silky Fowls Black-Bone Chicken is one of the excellent local chicken breeds in ancient China, which is loved by the public due to its rich melanin content. Research has shown that black-bone chicken melanin has functions such as nourishment, antioxidation, anti-aging, promoting body metabolism, maintaining internal environmental stability, and anti-mutation [1]. Therefore, in the breeding work of black-bone chicken, melanin content has also become an important indicator. Melanocytes are the main cells that synthesize and secrete melanin in animal skin and originate from the neural crest, and then gradually migrate to the eye, the epidermal basal layer of the skin and hair follicles, and are the cytological basis for hair and skin coloration in animals. The process of melanin production takes place in the melanocytes originating from the melanocytes of the nerve ridges and is extremely complex [2,3,4]. Melanin synthesis is influenced by many factors, including enzymes [5], hormones, cytokines [6], and inorganic ions [7], and changes in the physical and chemical environment around melanocytes can significantly affect the biosynthesis of melanin. Melanin synthesis is catalyzed by a variety of intracellular enzymes, but the most important is tyrosinase (TYR), the rate-limiting enzyme of melanin synthesis, whose expression and activity determine the rate and yield of melanin production and play a decisive role in the process of melanin synthesis [8]. The synthesis of melanin is also influenced by cytokines. For example, α-melanocyte-stimulating hormone (α-MSH), which increases intracellular cAMP content by binding to the corresponding receptor (MCIR) in melanocytes, increases tyrosinase activity, thus stimulating melanocyte differentiation and proliferation, and promotes melanin biosynthesis [9].

Melatonin (MT), also known as N-acetyl-5-methoxytryptamine, is an indole-like hormone widely distributed in microorganisms, plants, and animals. Initially, only the pineal gland was known to secrete melatonin in animals, and its function was limited to the regulation of individual circadian rhythms and seasonal signals. However, studies published in the following decades showed that most organs and tissues are capable of synthesizing melatonin, such as the retina, gastrointestinal tract, and gonads [10], and perform a variety of physiological functions, such as anti-tumor properties [11] and antioxidant functions [12]. In addition, many studies have found that melatonin affects melanin synthesis, but it plays a different role in different cells. It was found that melatonin reduces the phosphorylation of glycogen synthase kinase-3β, and pre-incubation with specific inhibitors of this protein kinase reduces the expression and activity of tyrosinase, blocks the PI3K/AKT pathway and stimulates melanin production in human SK-MEL-1 melanoma cells [13]. Moreover, melatonin triggered the appearance of premelanosomes and MNT-1 cells synthesize de novo endogenous melatonin, indicating that melatonin could also reduce melanogenesis in human MNT-1 melanoma cells [14]. However, the role of melatonin in chicken melanocytes has not been reported.

In this study, we isolated and characterized melanocytes, and detected the effect of melatonin on the synthesis of melanin, melanin synthesis-related genes, and tyrosinase activity in the melanocytes of Silky Fowls Black-Bone chicken. We also performed RNA-seq on melatonin-treated melanocytes to understand the downstream regulatory mechanism of melatonin. The results of our study will be helpful for understanding the mechanism of melatonin in primary melanocytes of Silky Fowls Black-Bone Chicken and provide a theoretical basis for changing melanin content in the breeding work of black-bone chickens.

## 2. Materials and Methods

### 2.1. Isolation and Culture of Melanocytes

Remove 20-day-old 20-day embryo age Silky Fowls Black-Bone chicken embryos and place them in a new culture dish. Under aseptic conditions, open the peritoneum and wash away the oil with preheated PBS containing a 2% penicillin and streptomycin mixture. Transfer the peritoneum to a small bottle and cut it into small pieces with scissors. Put the tissue pieces into a 15 mL centrifuge tube and add digestion solution (Diapase II and trypsin at a 1:1 volume ratio) at 37 °C for 90 min. Then, add an equal volume of culture medium to stop the digestion. After repeated aspiration and beating, filter the mixture successively through 200-mesh and 400-mesh filters, 1000 rpm/min, 10 min, and remove the supernatant. Gently resuspend the pellet in culture medium, centrifuge at 1000 rpm for 10 min, and remove the supernatant. Repeat this step 2–3 times. After adding the culture medium, count the cells and seed them at a concentration of 5 × 10^5^ cells/mL in a 25 cm^2^ cell culture flask. Place the flask in a 5% CO_2_, 37 °C cell incubator and change the medium after 24 h. Subsequently, change the medium every 2 days until the cells are fully confluent.

### 2.2. Indirect Immunofluorescence Assay

Seed the cells in a 12-well plate and when the cell fusion reaches 70–80%, remove the culture medium, and wash with PBS three times for 5 min each. Add 4% paraformaldehyde and fix it at room temperature for 30 min. Remove the liquid, and wash with PBS three times for 5 min each. Add 0.1% Triton X-100 and incubate at room temperature for 20 min. Remove the liquid, and wash with PBS three times for 5 min each. Add 1% bovine serum albumin and incubate at room temperature for 30 min. Remove the liquid and add 200 μL of the primary antibody (dilution ratio: 1:250). Incubate at 4 °C overnight. Remove the liquid, and wash with PBST three times for 5 min each. Add the secondary antibody (dilution ratio: 1:1000) and incubate at room temperature for 2 h. Remove the liquid, and wash with PBST three times for 5 min each. Add DAPI and incubate at room temperature for 10 min. Remove the liquid, and wash with PBST three times for 5 min each. Observe and capture images under an inverted fluorescence microscope. Primary antibodies include Rabbit anti-TYR, Rabbit anti-PAX3, Rabbit an-ti-S100 (Affinity Biosciences, Beijing, China), and Rabbit anti-MITF (Bioss Antibodies, Beijing, China). The secondary antibody was FITC Conjugated AffiniPure Goat Anti-Rabbit IgG (H + L) (Cambridge, UK).

### 2.3. Treat the Cells with Melatonin and CCK8 Assay

The cells were inoculated into 96-well plates. After the cells adhered to the wall, the old medium was removed and the medium containing different concentrations of melatonin (0, 10^−1^, 10^−2^, 10^−3^, 10^−4^, 10^−5^, 10^−6^, 10^−7^, 10^−8^, 10^−9^, 10^−10^, 10^−11^ mol/L) (Solarbio, Beijing, China) was added with 5 replicates in each group. The effect of melatonin on the cell viability of melanocytes was detected by CCK8 assay at 24 h, 48 h, and 72 h (Vazyme, Nanjing, China).

### 2.4. Total RNA Extraction, cDNA Synthesis, and Real-Time Quantitative PCR (RT-qPCR)

The cells were seeded into 6-well plates. After the cells adhered to the wall, the old medium was removed and the medium containing different concentrations of (0, 10^−4^, 10^−6^, 10^−8^ mol/L) melatonin was added. The cells were collected at 24 h, 48 h, and 72 h. Then, the total RNA was extracted using TRIzol Reagent (Vazyme, Nanjing, China) and stored at −80 °C. The cDNA synthesis of genes was performed using HiScript III RT SuperMix for qPCR (+gDNA Wiper) (Vazyme, Nanjing, China), and the expression levels of melanin synthesis-related genes were detected using the ChamQ Universal SYBR qPCR Master Mix (Vazyme, Nanjing, China). RT-qPCR was performed in the QuantStudio 5 real-time PCR instrument. Primer sets were designed with Primer 5.0 (Table 1), and synthesized by Tsingke Biotech (Beijing, China). Using chicken GAPDH reference genes. All assays were run in triplicate. Expression levels were quantified using the 2^−ΔΔCT^ method [15].

### 2.5. Detection of Melanin Content

The cells were seeded into 6-well plates. After the cells adhered to the wall, the old medium was removed and the medium containing different concentrations of (0, 10^−4^, 10^−6^, 10^−8^ mol/L) melatonin was added. The cells were collected at 24 h, 48 h, and 72 h and treated into cell suspension with trypsin. Based on previous research methods, we detected the melanin content in cells [16]. Firstly, we centrifuged the cell suspension at 4 °C and 1000 rpm for 10 min and discarded the supernatant. Then, we added 1 mL of 1 mol/L NaOH solution to the cells and placed them in an 80 °C water bath for 1 h. Lastly, we measured the absorbance at a wavelength of 475 nm to reflect melanin content.

### 2.6. Determination of Tyrosinase Activity

The cells were inoculated into 6-well plates. After cell adhesion, the old medium was removed and the medium containing the appropriate concentration of melatonin was added. The cells were collected at 24 h, 48 h, and 72 h, and treated into cell suspension with trypsin, and the supernatant was discarded. An amount of 90 μL of 1%Trition X-100 solution was added, and 10 μL of 0.1% L-DOPA (3,4-dihydroxyphenyl-L-ananine) (Solarbio, Beijing, China) was added to each well after 5 min of oscillation. Then, all the liquids were transferred to 96-well plates and incubated at 37 °C for 30 min. Absorbance was measured at a 450 nm wavelength.

### 2.7. RNA-seq

After the melanocytes were treated with 10^−4^ mol/L melatonin for 72 h, the cells were collected and sent to Novogene (Beijing, China) for library construction, sequencing, and bioinformatics analysis. Novogene uses standard extraction methods to extract RNA from cells, followed by strict quality control of the RNA samples. The mRNA with polyA tails is then enriched by Oligo (dT) beads, and the mRNA is randomly interrupted with divalent cations in NEB Fragmentation Buffer, and the library is built according to the Next^®^ Ultra™ RNA Library Prep Kit for Illumina ^®^(NEB, Beijing, China) for library construction. After the libraries were constructed, the libraries were initially quantified using a Qubit 2.0 Fluorometer and diluted to 1.5 ng/µL, followed by the detection of the insert size of the libraries using an Agilent 2100 bioanalyzer, and after the insert size met the expectations, the effective concentration of the libraries was accurately quantified by RT-qPCR. After the insert size met the expectation, RT-qPCR was performed to accurately quantify the effective concentration of the library (effective library concentration above 2 nM) to ensure the quality of the library. After passing the library inspection, the different libraries are pooled according to the effective concentration and the target downstream data volume required for Illumina sequencing. After quality control, we map cleaning readings to the Gallus gallus reference genome GRCg6a (GCA_000002315.5). The DEG analysis was performed between mock- and melatonin-treated cells using DESeq2 software(version 1.20.0) [17] under the condition of *p* value < 0.05 and |log2FoldChange| > 0. Subsequently, all DEGs were mapped to Gene Ontology (GO) terms [18], and gene numbers were calculated for every term. Significantly enriched GO terms (padj < 0.05) in DEGs compared to the genome background were defined by hypergeometric testing. Pathway enrichment analysis was also performed using the Kyoto Encyclopedia of Genes and Genomes (KEGG) database [19]. Pathways with padj < 0.05 were defined as significantly enriched pathways in DEGs.

### 2.8. Statistical Analysis

Statistical analyses of RT-qPCR data were performed using one-way ANOVA of SPSS 22.0 and GraphPad Prism 8 software (GraphPad Software Inc., San Diego, CA, USA) and presented as the mean ± SD. *p* < 0.05 (*) was considered to represent a statistically significant difference.

## 3. Results

### 3.1. Isolation and Characterization of Silky Fowls Black-Bone Chicken Melanocytes

To ensure that our isolated cells were melanocytes, we used a microscope to observe these cells and found cells began to stretch after fully adhering to the wall, showing bipolar dendrites or multipolar dendrites shapes (Figure 1A). In addition, it can be observed that there are melanin granules in the cells (Figure 1B). To further identify the isolated melanocytes, the cultured melanocytes were detected by an indirect immunofluorescence assay. The results showed that the expressions of MITF, S-100, PAX3, and TYR in isolated melanocytes were positive signals (Figure 1C).

### 3.2. Melatonin Inhibits Melanin Synthesis in Melanocytes

We treated melanocytes with different melatonin concentrations, then detected cell proliferation ability at 24 h, 48 h, and 72 h. CCK8 results showed that melatonin inhibited cell proliferation after a 48 h treatment of melanocytes (Figure 2A). From these different treatment concentrations, we screened three concentrations with better inhibitory effects (10^−4^, 10^−6^, and 10^−8^ mol/L) (Figure 2B), and these three treatment concentrations were also selected for subsequent experiments. Next, we examined the effects of melatonin on melanin synthesis in melanocytes from different aspects. First, we investigated the effects of melatonin treatment of melanocytes at different concentrations on the expression of genes related to melanin synthesis. After treatment of melanocytes with different concentrations of melatonin for 24 h (Figure 2C), 48 h (Figure 2D), and 72 h (Figure 2E), RT-qPCR results showed that melatonin inhibited the expression of these genes related to melanin synthesis (*MITF*, *MC1R*, *TYR*, *PMEL*, and *EDNRB2*) at all three treatment time points. Second, we directly detected changes in melanin content in melatonin-treated melanocytes. From the results, we can see that after 24 h, 48 h, and 72 h treatment of melanocytes with three different concentrations of melatonin, the content of melanin was significantly reduced (Figure 2F). Tyrosinase is the rate-limiting enzyme that regulates melanin production; so, we examined the effect of melatonin on tyrosinase activity in melanocytes. The results showed that melanocyte tyrosinase activity was reduced after 24 h, 48 h, and 72 h of melatonin treatment at different concentrations (Figure 2G). In summary, we found that melatonin inhibits melanin synthesis in melanocytes based on all these results.

### 3.3. Overview of RNA Sequencing Data, Differential Expression Gene Screening

To further understand the mechanism of the downstream regulation of melatonin, we constructed six transcriptome libraries from control and melatonin-treated cell samples to detect the gene expression profile between melatonin-untreated and -treated melanocytes. After quality control, the Q20 and Q30 percentages of clean data for all samples were more than 97% and 92%, respectively. The GC content of clean reads in each sample ranged from 50.44% to 51.35% (Appendix A). These findings indicated that the resulting clean reads were of high quality and suitable for further analysis.

After further analysis, the high-quality clean reads were subsequently mapped to the reference genome after removing the rRNA-mapped reads. As shown in Appendix A, more than 85.2% of the clean reads were successfully mapped to the reference genome, and more than 89% of the unique mapping reads were for six samples. In addition, 77.5~78.7% of the localized sequences were in exonic regions, 8.0~10.0% in intronic regions, and 12.4~13.4% in intergenic sequences (Appendix A). The distribution of gene expression in each sample is shown in Figure 3A.

The differentially expressed transcripts were screened by comparing the transcriptomic data of the control and melatonin-treated groups. A total of 1703 differentially expressed mRNAs were screened, respectively, under the condition of *p* value < 0.05 and |log2FoldChange| > 0, of which 779 mRNAs were significantly upregulated (about 45.7%) and 924 mRNAs were significantly downregulated (about 54.3%) (Figure 3B,C). The heat map of differentially expressed gene clustering is shown in Figure 3D.

### 3.4. Enrichment Analysis of Differentially Expressed Genes

For functional annotation and biological process analysis of the differentially expressed mRNAs, GO and KEGG functional enrichment analysis was performed on 1703 differentially expressed genes mentioned in Section 3.3. GO enrichment analysis revealed that these genes were enriched to terms such as ribosomes, ribosomal subunit, structural constituent of ribosome, mitochondrial protein complex et al. (Figure 4C). KEGG enrichment analysis revealed that these genes were enriched in the ribosome, mTOR signaling pathway, TGF signaling pathway, and other pathways, among which the mTOR signaling pathway was associated with melanin synthesis (Figure 4A). Further analysis of the transcriptome data found that the mTOR signaling pathway enriched by KEGG was downregulated. In addition, we also found the MAPK signaling pathway and Wnt signaling pathway in the downregulated signaling pathway, which are also related to melanin synthesis (Figure 4B). The result of the KEGG enrichment analysis prompted us to believe that melatonin may decrease melanin synthesis of melanocytes by inhibiting the mTOR signaling pathway.

### 3.5. Validation of DEGs by RT-qPCR

In this study, the expression of DEGs between the two groups was verified using RT-qPCR. Eight DEGs were randomly selected for RT-qPCR analysis to verify the expression profiles of DEGs generated using RNA sequencing: *CDK3*, *COX7C*, *MBL2*, *RPS21*, *SLC4A8*, *DLX3*, *GLRB*, and *SLC23A1*. As shown in Figure 5, the expression trend of the selected DEGs generated by RNA sequencing was consistent with the levels obtained using RT-qPCR, confirming that the data generated by RNA sequencing were reliable.

## 4. Discussion

The black-bone chicken has a high medicinal and nutritional value, and its melanin has strong in vitro antioxidant effects by scavenging DPPH radicals, superoxide anion radicals, and inhibiting lipid peroxidation. Melatonin is an indole-like hormone secreted by the pineal gland and is involved in important physiological processes such as the regulation of rhythm. In addition, several studies have found that melatonin affects melanin synthesis through different pathways in human melanoma cells. Therefore, the present study was conducted to investigate the effect of melatonin on melanin synthesis in melanocytes of Silky Fowls Black-Bone chickens.

We isolated melanocytes from 20-day embryo age Silky Fowls Black-Bone chicken embryo and characterized the isolated cells by various means such as cytomorphological observation and indirect immunofluorescence assay. The morphological characteristics of the cells are one of the important methods to identify the cell type. The first human melanocytes cultured by Eisinger were multipolar dendrites [20], while most of the mouse skin melanocytes cultured by Dorothy were bipolar dendritic [21]. Taihe Black-Bone chicken melanocytes cultured by Xiong Miao were also bipolar dendrites or multipolar dendrites [22]. Our cultured melanocytes also show bipolar dendrites or multipolar dendrites, which is consistent with the results of Xiong Miao. Melanocytes are also similar to those of other species, possibly due to differences that exist between species. An indirect immunofluorescence assay is a commonly used technique for cell identification. There are about 17 marker antibodies and antigens commonly used to identify melanocytes, four of which were selected for this study. S-100 is an acidic coelomic protein, which is derived from neural crest cells, and melanocytes are derived from the migration and differentiation of neural crest cells, which also contain S-100 protein inside the cells; so, the cultured cells can be identified as melanocytes by S-100 [23]. The PAX3 gene is first expressed in embryonic neural crest precursor cells and plays an important role in the differentiation and migration of melanoblasts into melanocytes during this process [24]. In addition, TYR is the rate-limiting enzyme for melanin synthesis, which is widely expressed and present in melanocytes [25]. And MITF regulates the expression of TYR and Karin used immunocytochemical staining of MITF protein to study the mechanism of melanin synthesis in melanocytes [26], which suggests that MITF is expressed in melanocytes. All four proteins were positive in this study, proving that the cells we isolated were chicken melanocytes.

In mammals, melatonin is mainly produced in the pineal gland and retina, with a small amount synthesized in the skin, and is considered a key neurohormone regulated by circadian rhythms [27,28]. It has many other biological functions [29,30] and has additionally been reported to affect melanin synthesis [31,32]. In most cells, melatonin inhibits melanin synthesis. For example, in human primary melanocytes, melatonin inhibits melanin synthesis via the p53-TYR pathway [33]. In human MNT-1 melanocytoma cells, melatonin exerts oncostatin capacity and decreases melanogenesis [14]. However, the melatonin metabolite 5-methoxy tryptamine (5MT) enhanced melanin pigmentation in human SKMEL-188 melanoma cells [34]. In the present study, melatonin inhibited melanin synthesis in chicken primary melanocytes, which is consistent with most of the results. In Yang’s results, melatonin treatment downregulated melanin synthesis-related genes such as *MITF*, *TYR*, and *TYRP1* in HaCaT cells [35], which is consistent with our results in which melatonin inhibited the expression of melanin synthesis-related genes (*MITF*, *TYR*, *MC1R*, *PMEL*, and *EDNRB2*). Our results showed that melatonin treatment inhibited the tyrosinase activity of chicken melanocytes, and in Sevilla’s results, either 100 μM or 200 μM melatonin inhibited human epidermal tyrosinase activity [36], and the two results were consistent. In conclusion, these results suggest that melatonin inhibits melanin synthesis in chicken primary melanocytes.

Next, to investigate how melatonin inhibits melanin synthesis, we performed RNA-seq in the hope of finding the signaling pathways involved in melatonin in chicken melanocytes. The KEGG results show that the top 20 most enriched signaling pathways included a pathway related to melanin synthesis, the mTOR signaling pathway. The KEGG results of the downregulated signaling pathways identified three signaling pathways related to melanin synthesis, namely the MAPK signaling pathway, the Wnt signaling pathway, and the mTOR signaling pathway. In Sevilla’s study, melatonin inhibited melanin synthesis through the PI3K/AKT signaling pathway [37]. In our study, KEGG was enriched to the mTOR signaling pathway, and the study demonstrated that the PI3K/AKT downstream target is the mammalian target of rapamycin protein (mTOR). And, it has also been demonstrated that melatonin acts through the PI3K/AKT/mTOR signaling pathway [38]. In human epidermal keratin-forming cells, isoliquiritigenin (ISL) effectively inhibited melanin synthesis, and the decrease in p-AKT and p-mTOR proteins after ISL treatment suggested that the PI3K/AKT/mTOR signaling pathway was involved in ISL-induced melanin degradation [39]. In conclusion, the KEGG results suggest that melatonin may be reducing melanin synthesis through the PI3K/AKT/mTOR signaling pathway.

## 5. Conclusions

In our study, we reported that melatonin could decrease the expression of melanocyte melanin synthesis-related genes (*MITF*, *MC1R*, *TYR*, *PMEL*, and *EDNRB2*), reduce melanin synthesis, and decrease tyrosinase activity. Meanwhile, we also found that some genes were enriched in the mTOR signaling pathway based on RNA-seq data, which prompted us that melatonin may affect the synthesis of melanin in primary melanocytes of Silky Fowls Black-Bone Chicken via the mTOR signaling pathway. These results reveal the important effect of melatonin in the synthesis of melanin in primary melanocytes of Silky Fowls Black-Bone Chicken, broadening the insight into the role of melatonin in primary melanocytes of Silky Fowls Black-Bone Chicken.

## Figures and Tables

**Figure 1 genes-14-01648-f001:**
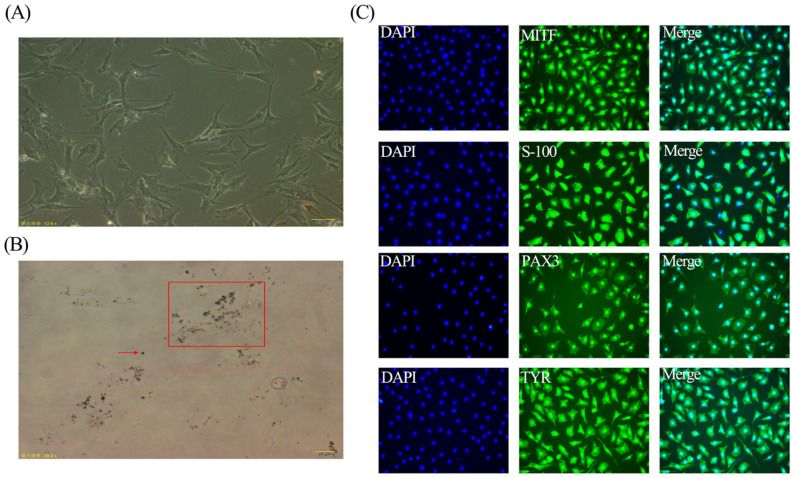
Morphological observation and identification of melanocytes. (**A**) Observation of culture of melanocytes (scale bar, 50 μm). (**B**) Melanin granules of melanocytes; the black dots in the red box and the black particles indicated by the red arrows in the image are both melanin granules (scale bar, 20 μm). (**C**) Identification of isolated melanocytes by IFA (blue fluorescence is DAPI, green fluorescence is MITF, S-1OO, PAX3, TYR; scale bar, 50 μm).

**Figure 2 genes-14-01648-f002:**
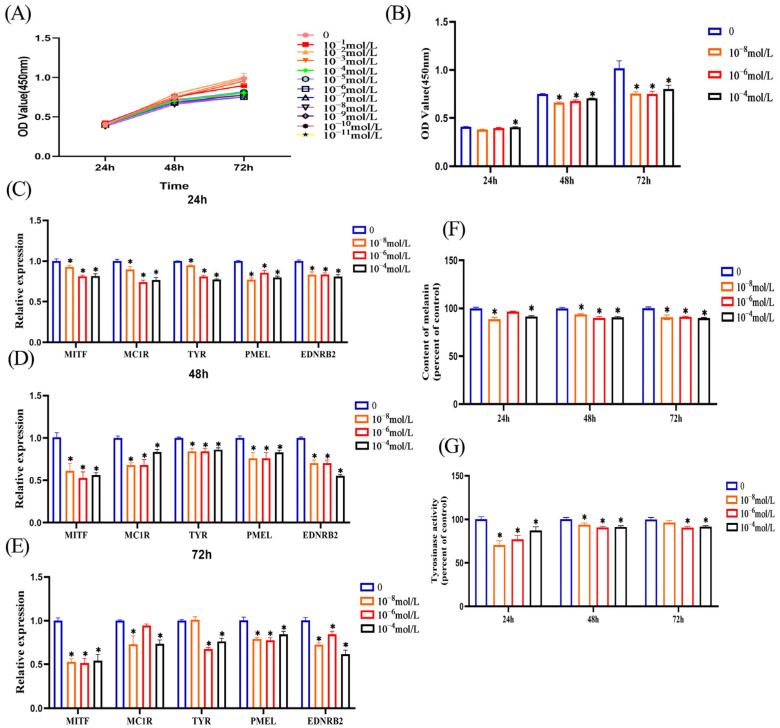
Melatonin inhibits melanin synthesis in melanocytes. (**A**) and (**B**) cell viability detected by CCK8 when melanocytes were treated with different concentrations of melatonin for 24 h, 48 h, and 72 h. RT-qPCR was used to detect the expression of melanin synthesis-related genes when melanocytes were treated with different concentrations of melatonin for 24 h (**C**), 48 h (**D**) and 72 h (**E**). (**F**) Detect the change in melanin content after melatonin treatment of melanocytes for 24 h, 48 h and 72 h by NaOH. (**G**) The tyrosinase activity of melanocytes after 24 h, 48 h and 72 h of melatonin treatment was detected by 0.1% L-DOPA. Data are expressed as means ± SD (n = 3). The one-way ANOVA was used for statistical significance (* *p* < 0.05).

**Figure 3 genes-14-01648-f003:**
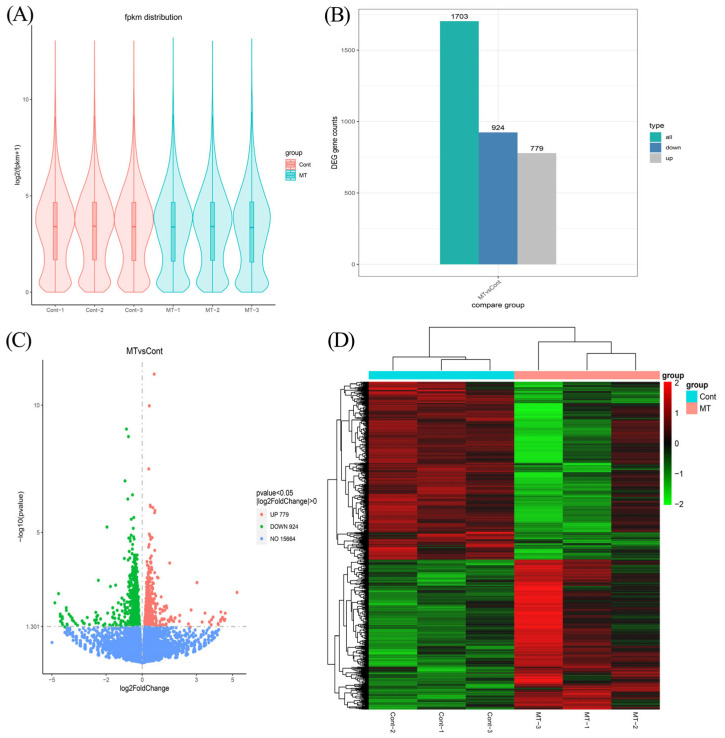
RNA sequencing results’ analysis. (**A**) Violin plot of gene expression patterns for each sample, with the middle horizontal line representing the median. (**B**) The numbers of upregulated and downregulated DEGs were evaluated. (**C**) Volcano plot showing the DEGs. The upregulated DEGs are represented by red dots, downregulated DEGs are represented by green dots, and genes with no significant differences in expression are represented by blue dots. (**D**) The heat map of differentially expressed gene clustering for each sample.

**Figure 4 genes-14-01648-f004:**
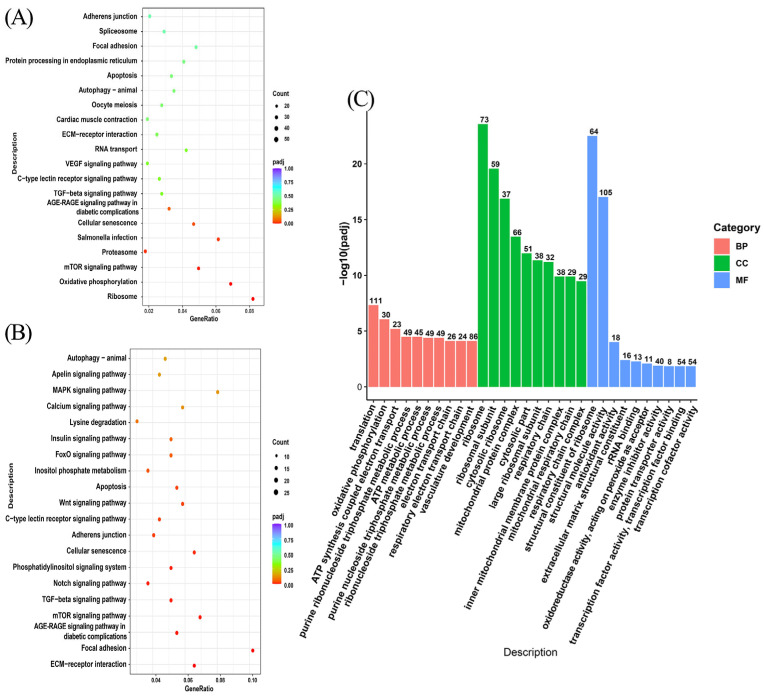
Enrichment analysis of differentially expressed genes. (**A**) The top 20 significantly enriched KEGG pathways. (**B**) The top 20 significantly enriched downregulated KEGG pathways. (**C**) The top 30 significantly enriched GO terms.

**Figure 5 genes-14-01648-f005:**
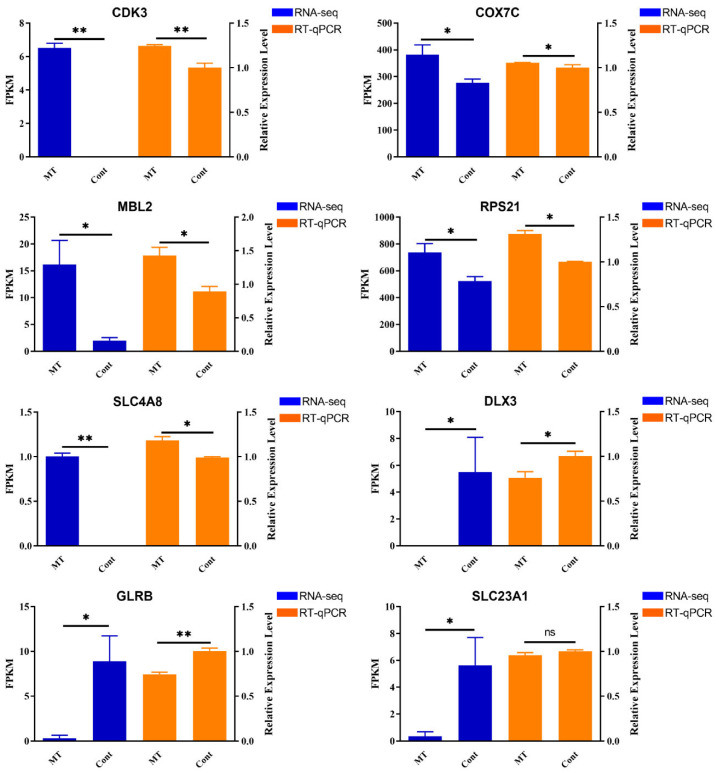
Confirmation of the transcriptome sequencing data by RT-qPCR. Data are expressed as means ± SD (n = 3). The one-way ANOVA was used for statistical significance. (ns: *p* > 0.05; * *p* < 0.05; ** *p* < 0.01).

**Table 1 genes-14-01648-t001:** The primer sequences for RT-qPCR.

Gene	Primer Sequence (5′-3′)	Product Size (bp)	Accession Number
*MITF*	F: TGTGACTGAACCAACTGGCACTTAC	157	ENSGALT00010053658.1
	R: GCTCCGCCTGCTACTCGTT	191	ENSGALT00000077972.3
*PMEL*	F: TTGTCTACGTGTGGTGGAC		
	R: CTGGTCGGTGATGCTGAACT		
*EDNRB2*	F: GAGGAAGTTTAATTCACTAGGAACC	104	ENSGALT00010029474.1
	R: TTGCTTGGGTCTTGGTCTGAT		
*TYR*	F: CACTCTTAGGTGGCTCCAATGTG	154	ENSGALT00010018719.1
	R: CAGTCCCAGTAGGGGATGGTGAA		
*MC1R*	F: GCCCTTCTTCTTCCACCTCAT	83	ENSGALT00010023383.1
	R: AGAGGTTGAAATAGCTGAAGAAGCA		
*CDK3*	F: ACTTGAAGCCACAGAACTTGC	251	ENSGALT00010071972.1
	R: ATCGATCTCAGAGTCCCCTT		
*SLC4A8*	F: TTGCCTACAAAGCCAAGGACCG	259	ENSGALT00010062208.1
	R: CGCTTCACATCCAGGATCAAACC		
*MBL2*	F: TCCTGCAGTCAATGGATTACCAG	127	ENSGALT00000143518.1
	R: TTTTAATCCTTGGGGTCCTG		
*RPS21*	F: GATCTCTACGTGCCTCGGAA	162	ENSGALT00010051720.1
	R: CCCCATCCTACGAATTGCT		
*COX7C*	F: TTCACTACCTCCGCCCTTCGT	109	ENSGALT00010014887.1
	R: AGAATGCACACATCATTGCCAGT		
*GLRB*	F: GACTCAAACTGCCCAACGA	219	ENSGALT00010019616.1
	R: TCAAGTCCAAAGGGCACGA		
*DLX3*	F: ATTACAGCGGCCAGCACGACT	243	ENSGALT00010063622.1
	R: TCCCGTTGACCATCCGCACCT		
*SLC23A1*	F: CACCCTCATCCAGACCACCGT R: ACCCAGTTGCCATAGATCTGCT	141	ENSGALT00010019860.1
*GAPDH*	F: CGATCTGAACTACATGGTTTAC	153	ENSGALT00000114062.1
	R: TCTGCCCATTTGATGTTGC		

## Data Availability

Data presented in this study are available upon request from the corresponding author.

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
