# Peer review of "Combine with RNA-seq Reveals the Effect of Melatonin in the Synthesis of Melanin in Primary Melanocytes of Silky Fowls Black-Bone Chicken"

_genes, 2023, doi:10.3390/genes14081648_

Round 1
Reviewer 1 Report
The authors have done extensive work on culturing melanocytes and studying melanin synthesis by melatonin exposure. Various methods and tools such as indirect immunofluorescence assay (IFA), qRT-PCR, RNA-seq and others were applied to evaluate genetic aspects in the melatonin activity of Silky Fowls Black-bone Chicken. The objectives of the experiment were achieved. However, the following notes need to be clarified.
You write: "The results of our study will be helpful to understand the mechanism of melatonin in primary melanocytes of Silky Fowls Black-bone Chicken ". What is your ultimate goal of studying this mechanism? Is it associated with the development of certain disorders and is it possible to influence melanin production through hormone therapy to inhibit the formation of cancer cells? Does this have a connection to the study of circadian rhythms? Or the study of this mechanism will help to influence phenotypic manifestations of traits, such as feather colouration or muscle colouration, etc.
Materials and Methods. What protocol did you use to isolate culture and inoculate melanocytes? Is this an approach developed in your laboratory or is it an established method (but who is the originator)?
Line 104, you exposed cells to different concentrations of melatonin. What is the possibility that chickens of the breed you are studying might have the same hormone levels outside of the experiment (e.g., 10-1, 10-2mol/L)?
In section "3.3 Overview of RNA sequencing data, differential expression gene screening" you display the data in the table for control and melanocyte-treated samples. The largest difference in GC-content between control and melanocyte-treated cells is 0.91% (51.23%, 51.12%, 51.35% - control, 50.44%, 51.00%, 50.56% - MT). Does this difference matter?
Could you please tell us more about what you wanted to show on the Violin plot, describe why it does not show differences in distributions of Control_1,2,3 and MT_1,2,3 samples?
In Figure 4 the inscriptions on the graphs are not visible enough, some of them are even indistinguishable, it would be better to make the font larger.
Author Response
Thank you very much for all the suggestions of this article, which have improved the manuscript. I have answered each question point-to-point. Thank you again for your work!
Question1: You write: "The results of our study will be helpful to understand the mechanism of melatonin in primary melanocytes of Silky Fowls Black-bone Chicken ". What is your ultimate goal of studying this mechanism? Is it associated with the development of certain disorders and is it possible to influence melanin production through hormone therapy to inhibit the formation of cancer cells? Does this have a connection to the study of circadian rhythms? Or the study of this mechanism will help to influence phenotypic manifestations of traits, such as feather colouration or muscle colouration, etc.
Answer 1: First, excessive melanin deposition induces melanoma in humans, rats, dogs, and other animals; in the present study, melatonin inhibited melanin deposition, providing evidence that melanin production is affected by hormone therapy, secondly, it has been demonstrated that biological rhythms play a role in regulating melanin deposition. In addition, black color is an economically important shape for the ebony chicken, and the study of the mechanism by which melatonin affects melanin synthesis by melanocytes provides a theoretical basis for the study of traits related to melanin deposition.( Slominski AT, Hardeland R, Reiter RJ. When the circadian clock meets the melanin pigmentary system. J Invest Dermatol. 2015 Apr;135(4):943-945. doi: 10.1038/jid.2014.553.)
Question2: Materials and Methods. What protocol did you use to isolate culture and inoculate melanocytes? Is this an approach developed in your laboratory or is it an established method (but who is the originator)?
Answer 2: The separation method of primary melanocytes used in this manuscript is based on Dr. Donghua Li's doctoral thesis, and of course has been optimized in some places.(Donghua Li. Identification and Regulation Mechanism of Candidate Genes for Melanin Deposition in Skin of Xichuan Black Bone Chicken[D]. Henan Agricultural University, 2019.)
Question3: Line 104, you exposed cells to different concentrations of melatonin. What is the possibility that chickens of the breed you are studying might have the same hormone levels outside of the experiment (e.g., 10-1, 10-2mol/L)?
Answer 3: We searched the relevant literature, so far, we have not found the literature to detect the melatonin level of Silky Fowls Black-bone Chicken, and for the melatonin concentration in this study, we are referring to the Kleszczynski K’s research, and we set up 11 added concentration, and finally screened the better effect of the 11 added concentration to be used in the subsequent experiments. Also, the cells used for the experiment were from the same batch of cells, so their initial hormone levels were the same; the only variable was the different concentrations of melatonin that were added.( Kleszczynski K, Kim TK, Bilska B, Sarna M, Mokrzynski K, Stegemann A, Pyza E, Reiter RJ, Steinbrink K, Bohm M, Slominski AT. Melatonin exerts oncostatic capacity and decreases melanogenesis in human MNT-1 melanoma cells. J Pineal Res 2019,67,e12610.)
Question4: In section "3.3 Overview of RNA sequencing data, differential expression gene screening" you display the data in the table for control and melanocyte-treated samples. The largest difference in GC-content between control and melanocyte-treated cells is 0.91% (51.23%, 51.12%, 51.35% - control, 50.44%, 51.00%, 50.56% - MT). Does this difference matter?
Answer 4: GC content refers to the proportion of the two bases, G and C, to the total bases. Second-generation sequencing platforms have more or less a certain sequencing bias, and the GC content can assist us in determining whether the sequencing process is sufficiently random. There is a theoretical value for the GC content of a general genome, for example, the GC content of the human genome is usually around 40%. Therefore, if the map of GC content is found to deviate significantly from the theoretical value, it indicates that there is a high sequence bias in the sequencing process, and as a result, the chances of certain specific regions in the genome being sequenced repeatedly become higher, and the sequencing depth of these regions is much higher than the average sequencing depth, which will affect downstream variant detection and CNV analysis. Differences between samples are unimportant as long as they are within a reasonable range.
Question 5: Could you please tell us more about what you wanted to show on the Violin plot, describe why it does not show differences in distributions of Control_1,2,3 and MT_1,2,3 samples?
Answer 5: Violin plots are generally used to visualize the abundance expression of genes, which can show the distribution state of the data as well as the probability density. The graphs are interpreted as follows: â‘ The middle horizontal line represents the median Q2 (i.e., half of the data is greater than the median and above it, and the other half is less than the median and below it); â‘¡ The rectangle is the range from the lower quartile to the upper quartile; the upper edge of the rectangle is the upper quartile Q3, which represents that a quarter of the data is greater than the upper quartile, and the lower edge is the lower quartile Q1, which represents that a quarter of the data has a number is smaller than the lower quartile, the interquartile spacing IQR (upper quartile and lower quartile spacing); â‘¢ length represents the dispersion and symmetry of the non-anomalous data, long is dispersed, short is concentrated; the line running up and down through the violin graph represents the interval from the minimum non-anomalous value min to the maximum non-anomalous value max, and the lower and upper ends of the line represent the upper limit and the lower limit, respectively, and exceeding the range of the anomalous data; â‘£ the external shape of the rectangle For kernel density estimation, the length of the vertical axis of the graph represents the degree of data dispersion, and the length of the horizontal axis represents how much data is distributed in a certain vertical coordinate position.
The violin plot shows the distribution of the data and it does not necessarily show differences.
Question 6: In Figure 4 the inscriptions on the graphs are not visible enough, some of them are even indistinguishable, it would be better to make the font larger.
Answer 6: Figure 4 has been modified to enlarge the text inside the image, in addition, the modified image has replaced the original image.
Where revisions have been made in the manuscript have been are highlighted in yellow.
Reviewer 2 Report
1. General comments
Yang et al. reports the article, entitled 'Combine with RNA-seq reveals the effect of melatonin in the 2 synthesis of melanin in primary melanocytes of Silky Fowls 3 Black-bone Chicken' for publication in Genes journal. They isolated melanocytes from Silky Fowls Black-bone chicken embryos and treated them with different concentrations of melatonin, to explore the effect of melatonin on cell viability and melanin pigmentation in melanocytes, and further performed RNA-sequencing analysis with the aim of revealing the mechanism. From the previous, it is well known that melatonin may inhibit melanin pigmentation in various ways, i.e., inhibition of the group of melanogenic genes, paracrine effect, etc. In this regard, the specific aims of this study are not clear. Why did the authors use Silky Fowls 3 Black-bone chicken? In addition, no information on this chicken has been included in the introduction.
-The results from cultured melanocytes from this chicken are not solid and not convincing (Figure 2). Melatonin measurement data from the cell culture are not found, even procedure to measure it is presented (Line 122~).
The effects of melatonin on cell viability and alteration of transcriptions of melanonogenic genes are marginal, so it is hard to judge the effects of melatonin. How was the concentration of 10^-4 mole/L chosen? What are the criteria for choosing it? The authors need to explain it.
- There should be a statistical analysis for the validation research. Even if they go in the same direction, it doesn't seem convincing. Particularly RNA-seq results show significant variation between samples.
-Table 2 and 3 can be removed and included as supplementary tables.
-Calculationof differential gene expression with RT-qPCR analysis procedure and reference are missing.
2. Specific comments
Line 101~: No specific infomation of melatonin has bee included. Please add vendor, city, country.
Line 107: Real-time quantitative PCR(qRT-PCR)-->Real-time quantitative PCR(RT-qPCR)
Line 111: Please add information on Trizol.
Line 134: Please the full name of L-DOPA.
Line 148, 149, 162, 192, 265, 267: qRT-PCR should be changed to RT-qPCR.
Line 269~272: Except RPS21, DLX3, and GLRB, RT-qPCR results of the genes showed no or marginal difference.
Author Response
Thank you very much for all the suggestions of this article, which have improved the manuscript. I have answered each question point-to-point. In addition, the English in this manuscript has been improved. Thank you again for your work!
Question 1: Why did the authors use Silky Fowls Black-bone chicken? In addition, no information on this chicken has been included in the introduction.
Answer 1: Silky Fowls Black-bone Chicken is one of the ancient excellent local chicken breeds in China, with a unique body shape and appearance. Silkie silky chicken is one of the ancient excellent local chicken breeds in China, with a unique body shape and appearance. Black-bone chicken is a kind of traditional Chinese medicinal material, and its meat, bones, viscera, blood, etc. can be formulated into compound prescriptions. Modern research has also confirmed that black-bone chicken melanin has the functions of nourishing, anti-oxidation, anti-aging, promoting body metabolism, maintaining internal environment stability and anti-mutation. The nutritional nourishment and medicinal value of black-bone chicken are closely related to melanin. Additionally, an introduction to Silky Fowls Black-bone Chicken has also been added in the Introduction section of the manuscript.
Question 2: The results from cultured melanocytes from this chicken are not solid and not convincing (Figure 2). Melatonin measurement data from the cell culture are not found, even procedure to measure it is presented (Line 122~).
Answer 2: The primary melanocytes used in this study were isolated from silky chicken embryos. According to relevant literature, chicken melanocytes are all isolated from silky chicken embryos, except for the different parts of separation. So the results of the studies performed on our isolated melanocytes are reliable. ( DP Han,SX Wang, YY Zhang, et al. Culture and Identification of Primary Melanocytes from Silky Fowl in vitro[J]. Acta Veterinaria et Zootechnica Sinica, 2015, 46(6):949-956. ;Donghua Li. Identification and Regulation Mechanism of Candidate Genes for Melanin Deposition in Skin of Xichuan Black Bone Chicken[D]. Henan Agricultural University, 2019.; Miao Xiong. Effect ofα-melanocyte stimulating hormone and cyclic adenosine monophosphate on proliferation and melanin synthesis of melanocytes from Taihe silky fowls in vitro[D]. Jiangxi Agricultural University, 2014.)
Regarding melatonin levels, melanocytes were isolated from multiple embryos and they were pooled so that the initial melatonin levels were consistent. In addition, melanin was detected in this study, not melatonin, and this study did not involve the detection method of melatonin.
Question 3: The effects of melatonin on cell viability and alteration of transcriptions of melanonogenic genes are marginal, so it is hard to judge the effects of melatonin. How was the concentration of 10^-4 mole/L chosen? What are the criteria for choosing it? The authors need to explain it.
Answer 3: In Kleszczynski K 's study, the effect of melatonin on the cell viability of MNT-1 melanoma cells was also marginal, so the marginal effect of melatonin on the viability of the cells can be explained. Second, melatonin significantly inhibited the expression of genes related to melanin synthesis. In addition, the concentration of melatonin used in this study is based on Kleszczynski K 's research, and our results are consistent with his results. ( Kleszczynski K, Kim TK, Bilska B, Sarna M, Mokrzynski K, Stegemann A, Pyza E, Reiter RJ, Steinbrink K, Bohm M, Slominski AT. Melatonin exerts oncostatic capacity and decreases melanogenesis in human MNT-1 melanoma cells. J Pineal Res 2019,67,e12610.)
In this study, we set a total of 11 melatonin treatment concentrations. We treated primary melanocytes with different concentrations of melatonin and tested the cell viability. We found that using 10^-4mol/L, 10^-6mol/L and 10^-8mol/L melatonin has a better effect on inhibiting cell viability, so these three concentrations were finally screened and used in subsequent studies.
Question 4: There should be a statistical analysis for the validation research. Even if they go in the same direction, it doesn't seem convincing. Particularly RNA-seq results show significant variation between samples.
Answer 4: We performed statistical analysis on the results of both RNA-Seq and RT-qPCR, and most of the results were statistically different. The large difference between samples of RNA-Seq indicates that the biological repetition is not good, which will affect the results of RNA-Seq to a certain extent, but the difference between the treatment groups is significant, indicating that the results of RNA-Seq are still reliable .
Question 5: Table 2 and 3 can be removed and included as supplementary tables.
Answer 5: Tables 2 and 3 are provided as supplementary tables in the manuscript..
Question 6: Calculationof differential gene expression with RT-qPCR analysis procedure and reference are missing.
Answer 6: The corresponding information has been added in the appropriate places in the manuscript.( Livak, K.J.; Schmittgen, T.D. Analysis of relative gene expression data using real-time quantitative PCR and the 2(-Delta Delta C(T)) Method. Methods 2001, 25, 402–408.)
Question 7: Line 101~: No specific information of melatonin has been included. Please add vendor, city, country.
Answer 7: The corresponding information has been added in the appropriate places in the manuscript.
Question 8: Line 107: Real-time quantitative PCR(qRT-PCR)-->Real-time quantitative PCR(RT-qPCR). Line 148, 149, 162, 192, 265, 267: qRT-PCR should be changed to RT-qPCR.
Answer 8: Changes have been made to the manuscript in appropriate places.
Question 9: Line 111: Please add information on Trizol.
Answer 9: The corresponding information has been added in the appropriate places in the manuscript.
Question 10: Line 134: Please the full name of L-DOPA.
Answer 10: The full name of L-DOPA has been added in the appropriate places in the manuscript.
Question 11: Line 269~272: Except RPS21, DLX3, and GLRB, RT-qPCR results of the genes showed no or marginal difference.
Answer 11: Statistical analysis has been performed on these 8 genes, except for SLC23A1, the expression levels of the remaining 7 genes are significantly different.
Where revisions have been made in the manuscript have been are highlighted in yellow.
Reviewer 3 Report
The paper: " Combine with RNA-seq reveals the effect of melatonin in the synthesis of melanin in primary melanocytes of Silky Fowls Black-bone Chicken" describes an in vitro experiment in which primary melanocytes were stimulated by melatonin. The effect of this stimulation on cell viability, the content of melanine, tyrosine activity and expression of genes related to melanin synthesis were evaluated. The paper is interesting. However, it is written very uncarefully, with very poor English. Moreover, several issues should be clarified
line 60-62 - Please extend this fragment
Material and Methods
Explain why an embryo peritoneum was used as a source of the melanocytes. Was there any medium stimulating melanocyte growth used during the in vitro culture?
2.1, 2.2 Please change the tense.
line 161statistical analysis - was the data normally distributed - if not, you should use a non-parametric test
Results:
line 172 - please explain IFA
line 184Please explain CCK8
line 193 - Gene names should be written in italics
Figure 2 Please provide the number of biological replicates or technical replicates. Were these the same samples as the samples used for RNA-seq?
Table 3 Please correct: "Intervention"
line 253 What was the input for the functional annotation? Did you analyse upregulated and downregulated genes separately?
Were the replicates for RNA-seq biological or technical replicates?
Figure 5 The data presented do not really support the validation of RNA-seq data.
.
Author Response
Thank you very much for all the suggestions of this article, which have improved the manuscript. I have answered each question point-to-point. Thank you again for your work!
Question 1: line 60-62 - Please extend this fragment.
Answer 1: Thank you for your comments. This fragment was extended in manuscript
Question 2: Explain why an embryo peritoneum was used as a source of the melanocytes. Was there any medium stimulating melanocyte growth used during the in vitro culture?
Answer 2: Connective tissues such as periosteum and peritoneum, both in the embryonic stage and during growth and development, show a large amount of melanin deposition. Therefore, it is a good choice to select periosteum and peritoneum for the isolation and culture of melanocytes.( DP Han,SX Wang, YY Zhang, et al. Culture and Identification of Primary Melanocytes from Silky Fowl in vitro[J]. Acta Veterinaria et Zootechnica Sinica, 2015, 46(6):949-956.)
The medium used for melanocytes was Melanocyte Medium (ScienCell, Cat. No. 2201), which contains essential amino acids, non-essential amino acids, vitamins, organic compounds, inorganic compounds, hormones, growth factors, trace minerals, and a low concentration of fetal bovine serum (0.5%).
Question 3: 2.1, 2.2 Please change the tense.
Answer 3: Changes have been made in the manuscript.
Question 4: line 161statistical analysis - was the data normally distributed - if not, you should use a non-parametric test.
Answer 4: The data conforms to a normal distribution. An example is shown in the figure below.
Question 5: line 172 - please explain IFA, line 184 Please explain CCK8.
Answer 5: The indirect immunofluorescence assay(IFA) is a commonly used immunological assay, the basic principle of which is to detect the presence of a target antigen by means of a fluorescently labeled secondary antibody, taking advantage of the specificity of antibody-antigen binding.
Cell Counting Kit-8 (CCK-8) can be used for simple and accurate analysis of cell proliferation and toxicity. The basic principle is that the reagent contains WST-8 [Chemical name: 2-(2-methoxy-4-nitrophenyl)-3-(4-nitrophenyl)-5-(2,4-disulfophenyl)-2H-tetrazole monosodium salt]. In the presence of the electron carrier 1-methoxy-5-methylphenazinium dimethyl sulfate (1-Methoxy PMS), it is reduced by dehydrogenase in the cell to a highly water-soluble yellow formazan product (Formazan dye). The amount of formazan produced is proportional to the number of living cells. This property can therefore be utilized for direct cell proliferation and toxicity analysis.
Question 6: line 193 - Gene names should be written in italics.
Answer 6: All the gene names in the manuscript have been written in italics.
Question 7: Figure 2 Please provide the number of biological replicates or technical replicates. Were these the same samples as the samples used for RNA-seq?
Answer 7: The number of biological replicates for CCK8 is 10, and the number of biological replicates for qRT-PCR is 3 and the number of technical replicates for qRT-PCR is 3, and the number of biological replicates for melanin content test is 6, and the number of biological replicates for tyrosinase activity is 6. The samples used for these experiments are not the same samples used for RNA-Seq.
Question 8: Table 3 Please correct: "Intervention".
Answer 8: "Intervention" has been revised to "Intergenic" in the manuscript.
Question 9: line 253 What was the input for the functional annotation? Did you analyse upregulated and downregulated genes separately.
Answer 9: GO and KEGG analysis was performed on 1703 differentially expressed genes mentioned in section 3.3. In addition, this study is concerned with what signaling pathway melatonin is involved in to inhibit melanin synthesis by melanocytes, and is not designed to find differentially expressed genes, so differentially expressed genes were not analyzed in this study.
Question 10: Were the replicates for RNA-seq biological or technical replicates?
Answer 10: The replicates for RNA-seq were biological replicates
Question 11: Figure 5 The data presented do not really support the validation of RNA-seq data.
Answer 11: The trend of quantitative PCR results was consistent with that of RNA-Seq results, indicating that the results of RNA-Seq were reliable.
Where revisions have been made in the manuscript have been are highlighted in yellow.

Round 2
Reviewer 2 Report
1. General comments
-Even extensive revisions have been made, still current version is not satisfactory for the publication.
- No scientific questions for this study have been proposed in the Introduction. Why is it necessary to investigate the pigmentation of melanin and what are the justifications. Is it for the poultry industry's needs, such as breeding, or is it for consideration as an animal model for melanoma research?
-Please replace qRT-PCR with RT-qPCR in Figure 5.
2. Specific comments
- Line 36-44: Please include the references.
-Please list the gene ID from Ensembl in Table 1 along with the product size for each PCR product.
- Are the data presented as the mean +/- SD or SEM in the legend of Figure 2? Please include the number of replicates as well.
-Line 137-143: It appears that home-made chemicals or a commercial kit are used to quantify melanin. There is need to specify it and and include necessary chemicals and procedures with a reference.
-Line 153 ~: Please include 1) the reference genome details (version) and 2) references or websites for the bioinformatics tools for FPKM, DESeq2, GO analysis, and KEGG pathways. It's also important to refer to the visualization tool.
In Figure 5, please add more details of legend, such as statistics presenting what * means. COX7C and SLC4A8 expressions by RT-qPCR are still questionable.
Author Response
Thank you very much for all the suggestions for this article, which have improved the manuscript. I have answered each question point-to-point. In addition, the English in this manuscript has been improved. Thank you again for your work!
Question 1: Even extensive revisions have been made, still current version is not satisfactory for the publication.
Answer 1: Thank you for your suggestion, we have made further revisions to the manuscript.
Question 2: No scientific questions for this study have been proposed in the Introduction. Why is it necessary to investigate the pigmentation of melanin and what are the justifications. Is it for the poultry industry's needs, such as breeding, or is it for consideration as an animal model for melanoma research?
Answer 2: We have made modifications to the introduction section in the manuscript (Lines 36-41 and 78-80).
Question 3: Please replace qRT-PCR with RT-qPCR in Figure 5.
Answer 3: We have made modifications to Figure 5 in the manuscript (Figure 5).
Question 4: Line 36-44: Please include the references.
Answer 4: We have added relevant references in the revised manuscript (Lines 37-40).
Question 5: Please list the gene ID from Ensemble in Table 1 along with the product size for each PCR product.
Answer 5: We have added relevant information to the revised manuscript (Table 1).
Question 6: Are the data presented as the mean +/- SD or SEM in the legend of Figure 2? Please include the number of replicates as well.
Answer 6: We have added relevant information to the revised manuscript (Lines 230-231).
Question 7: Line 137-143: It appears that home-made chemicals or a commercial kit are used to quantify melanin. There is need to specify it and include necessary chemicals and procedures with a reference.
Answer 7: We have made modifications to the methods section in the manuscript (Lines 141-145).
Question 8: Line 153 ~: Please include 1) the reference genome details (version) and 2) references or websites for the bioinformatics tools for FPKM, DESeq2, GO analysis, and KEGG pathways. It's also important to refer to the visualization tool.
Answer 8: We have added relevant information and references in the revised manuscript (Lines 171-178).
Question 9: In Figure 5, please add more details of legend, such as statistics presenting what * means. COX7C and SLC4A8 expressions by RT-qPCR are still questionable.
Answer 9: We have added relevant information to the revised manuscript. Lines 289-291(Figure 5). We checked the RT-qPCR data of COX7C and SLC4A8 and ensured that our data was accurate and there were significant differences between the two treatment groups.
